# OVT-B: A New Large-Scale Benchmark for Open-Vocabulary Multi-Object Tracking

**Haiji Liang**[1]**, Ruize Han**[2,3*]

[1]School of Software Technology, Zhejiang University
[2]Shenzhen Institute of Advanced Technology, Chinese Academy of Sciences
[3]Department of Computer Science, City University of Hong Kong
coolsea@zju.edu.cn, rz.han@siat.ac.cn

## Abstract

Open-vocabulary object perception has become an important topic in artificial intelligence, which aims to identify objects with novel classes that have not been seen during training. Under this setting, open-vocabulary object detection (OVD) in a single image has been studied in many literature. However, open-vocabulary object tracking (OVT) from a video has been studied less, and one reason is the shortage of benchmarks. In this work, we have built a new large-scale benchmark for open-vocabulary multi-object tracking namely OVT-B. OVT-B contains 1,048 categories of objects and 1,973 videos with 637,608 bounding box annotations, which is much larger than the sole open-vocabulary tracking dataset, *i.e.*, OV-TAO-val dataset (200+ categories, 900+ videos). The proposed OVT-B can be used as a new benchmark to pave the way for OVT research. We also develop a simple yet effective baseline method for OVT. It integrates the motion features for object tracking, which is an important feature for MOT but is ignored in previous OVT methods. Experimental results have verified the usefulness of the proposed benchmark and the effectiveness of our method. We have released the benchmark to the public at `https://github.com/Coo1Sea/OVT-B-Dataset`.

## 1   Introduction

Multi-object tracking (MOT) has achieved significant progress in tracking specific categories such as humans and vehicles [1, 2]. However, the classical MOT task mainly focuses on tracking targets of people and vehicles, which makes the methods encounter difficulties when extended to a broader range of target categories, limiting their practical application value. Recent studies in the open vocabulary detection (OVD) domain, such as  [3], [4], have demonstrated the capability to detect objects of categories unseen during training directly. As we all know, detection is the fundamental task of MOT. So the burgeoning OVD can greatly bring generalization ability to MOT models.

OVTrack [5] is the first study attempting to combine open vocabulary with multi-object tracking. This work develops a simple baseline method namely OVTrack by combining a state-of-the-art open vocabulary detector and an appearance matching-based track head. As the basic study, based on a large generic MOT dataset TAO, it builds a new dataset containing OV-TAO-val and OV-TAO-test for performance evaluation, with a comprehensive tracking metric TETA[6] as the evaluation criterion.

Open-vocabulary multi-object tracking (OVMOT), as a new yet practical task, is very promising with great research potential. However, the largest challenge for the further study of this topic is the lack of a comprehensive benchmark. We all know that accurately evaluating the performance of new OVMOT methods requires a benchmark dataset containing *a large number of videos and categories*.

---

*Corresponding Author.

38th Conference on Neural Information Processing Systems (NeurIPS 2024) Track on Datasets and Benchmarks.

However, previous research utilized modified OV-TAO-val and OV-TAO-test as insufficient evaluation datasets. As stated in [7] – 'A more dynamic, challenging video dataset is needed to fully explore the potential of vision language models for open vocabulary learning.'

First, in terms of the object categories, although the basic dataset TAO has a large number of 833 categories, OV-TAO-val includes 330 object categories, composed of 295 base classes and only 35 novel classes. This is because, following the base/novel category setting in the LVIS [8] dataset, the authors in [5] use the overlapped base/novel categories between LVIS and TAO to build OV-TAO-val. This number of categories does not meet the large vocabulary evaluation needs in open vocabulary research. Especially the amount of new classes especially hardly reflects the model's true open vocabulary tracking capability. The category setup and data distribution of the OV-TAO-test dataset is almost identical to the OV-TAO-val dataset, failing to serve as a practicable benchmark. Second, the annotation rate of the original TAO is sparse. Specifically, the frame rate of the videos in TAO is 30 fps, but the annotation is only 1 fps. Also, in each video, the objects are not very dense with up to 10 annotated targets per frame. Therefore, in this work, we build OVT-B, a large-scale **O**pen-**V**ocabulary multi-object **T**racking **B**enchmark containing 1,973 videos and 637,608 annotated objects from 1,048 categories, as shown in Figure 1, surpassing the diversity of all current MOT datasets. Besides the diverse categories and large scale, the target presence and annotation density also exceed those of existing OV-TAO-val and OV-TAO-test datasets. OVT-B also includes some attributes especially for the MOT task, *e.g.*, the scenarios of out-of-view, fast motion, mutual occlusion, and objects with various sizes, shapes, *etc*.

We also develop a simple baseline method for open-vocabulary tracking. Specifically, the existing method OVTrack [5] solely relies on appearance features for object association, neglecting motion information, which is also a strong cue in the classical MOT task. This way, in this work, we propose OVTrack+, a simple method that combines appearance and motion information for association, enhancing performance by integrating the motion cues.

We summarize the main contributions of this work. We construct OVT-B, the first benchmark specifically designed for the open-vocabulary multi-object tracking (OVMOT) task, which is a massive and richly categorized dataset, with dense objects and full annotations. OVT-B can better provide a new platform for the research and evaluation of OVMOT. We also develop a new baseline method OVTrack+, which exploits the potentialities of motion features for OVMOT. We also conduct extensive experiments on OVT-B and report the comparison results of a series of approaches of OVMOT. Through the above effort, we provide a benchmark study for promoting the development of OVMOT.

## 2 Related Work

**Multiple Object Tracking.** Multi-object tracking (MOT) aims to detect, classify, and associate multiple object targets within video sequences. The classical MOT paradigm, *Tracking by Detection*, initially employs detectors to identify objects in frames, followed by trackers to associate these objects. Predominantly, this framework focuses on target association using two distinct types of cues. A notable example of using location and motion cues is SORT [1], which leverages Kalman filtering [9] to forecast trajectories and the Hungarian algorithm [10] for matching detection results with these trajectories. On the other hand, some works utilize appearance cues (re-identification, *ReID* methods) for target association, such as POI [11] and DeepSORT [2]. Recent researches suggest that combining both cues can achieve better performance, as demonstrated by Deep OC-SORT [12], BoT-SORT [13], and StrongSORT [14]. An alternative mainstream paradigm is *Joint Detection and Tracking*, which primarily explores the synergy between detection and tracking tasks. JDE [15] and FairMOT [16] are exemplary works that integrate detection with appearance embeddings extraction in one stage. Another way incorporates object displacement prediction into the detector, as illustrated by D&T [17], Tracktor [18], and CenterTrack [19]. Some approaches introduce the Transformer [20] architecture into MOT, aiming to model the tracking task through learning deep representations, such as MOTR [21] and TrackFormer [22]. However, these end-end methods often fall short of two-stage methods in terms of accuracy. Furthermore, several studies adopt offline methods, treating target association as a global optimization challenge across the entire sequence and employing graph-based models or graph neural networks [23, 24].

**Open-Vocabulary Object Detection.** To address the challenge of detecting novel category objects, researchers have proposed three approaches: Open-Set/Open World/Out-Of-Distribution (OOD) Learning, Zero-Shot Learning, and Open Vocabulary Learning. In Open-Set/Open World/OOD Learning, models need to recognize objects of novel classes and classify them as *unknown*. In zero-shot learning, models need to classify the novel categories with additional knowledge. In open vocabulary learning, which has become increasingly mainstream, models are allowed to classify novel categories using low-cost knowledge sources. The foundational assumption of open vocabulary learning is access to large-scale image captions available in network data. Based on this, OVR-CNN [25] first introduced the concept of open vocabulary object detection (OVOD), utilizing image captions to gain additional knowledge. Then, the contrastive learning model CLIP [26], leveraging the abundance of image-text pairs on the web, became a secondary source of knowledge for OVOD. Given the extensive knowledge of Visual Language Models (VLMs) like CLIP, employing a VLM to train a detector head is an intuitive approach. VilD [3] pioneered the use of the Knowledge Distillation method to build an OVOD model. Inspired by the DETR [27] series, OV-DETR [28] was introduced, innovating the original matching mechanism. The idea of aligning region and text also influenced subsequent works such as BARON [4] and VLDet [29]. Moreover, two knowledge sources have been further explored, i.e., pseudo labeling, with VL-PLM [30] and RegionCLIP [31] as exemplary works, and prompting engineering, as demonstrated by DetPro [32] and PromptDet [33]. Recently, the trend has shifted towards more innovative pre-training methods, e.g., CFM-ViT [34] and CoDet [35].

**Open-World and Open-Vocabulary MOT.** In traditional MOT benchmarks such as KITTI [36], MOTChallenge [37, 38], and DanceTrack [39], the categories are typically restricted to humans or vehicles. Many MOT algorithms achieve higher accuracy by leveraging the prior knowledge associated with these categories. As a result, the majority of MOT models cannot track objects across more general categories. However, there remains significant interest in tracking objects beyond humans and vehicles, such as wildlife (e.g., bats [40] and bees [41]). In response to this need, BLP [42] introduced the concept of Generic Multiple Object Tracking (GMOT), which expands the task from tracking multiple objects within some categories to some generic categories. Subsequently, GMOT-40 [43] established a new benchmark for this task, defining ten generic categories within its dataset. Besides, TAO [44] presents a broader variety of categories than previous datasets, creating a long-tail distribution that encourages the development of models capable of tracking more categories. With the popularity of Open-Set/Open World/OOD Learning, Open World Tracking [45] was introduced, evaluating the task of tracking *unknown* categories using OWTA as the evaluation metric. Following this development, SimOWT [46] achieved state-of-the-art performance in open-world tracking using a self-training paradigm. However, the capability for accurate classification should not be underestimated in tracking models. Therefore, OVTrack [5] presented the first framework for open vocabulary multiple object tracking, employing diffusion models to generate object pairs for training the model in association capabilities. Furthermore, VOVTrack [47] proposes to train the network using the (raw) videos rather than the image pairs in [5] in a self-supervised manner. To promote the development of this new and important topic, in this work, we complement the OVMOT evaluation benchmark and present a simple baseline.

Table 1: Statistics of MOT datasets and OVMOT datasets. We provide the number of classes (#Cls.), videos (#Vid.), tracks (#Track), boxes (#Box), frames (#Frm.) in these datasets, and the image resolution (Res./p), duration (Dur./second), and average number of objects per frame (#Obj.) and the annotation frame rate (Ann./fps).

| Datasets | #Cls. | #Vid. | #Track | #Box | #Frm. | Res. | Dur. | #Obj. | Ann. |
|---|---|---|---|---|---|---|---|---|---|
| MOT17 | 1 | 42 | 3993 | 901K | 33K | 480-1080 | 17-85 | 1-63 | 30 |
| MOT20 | 1 | 8 | 3833 | 2102K | 13K | 880-1080 | 17-133 | 1-94 | 30 |
| KITTI | 5 | 50 | 2600 | 80K | 15K | 512 | 20-90 | 0-30 | 10 |
| DanceTrack | 1 | 100 | 990 | 877K | 105K | 720-1080 | 20-108 | 1-22 | 20 |
| UAVDT | 3 | 100 | 2700 | 841K | 80K | 540-1080 | 3-99 | 1-122 | 6 |
| TAO | 833 | 2907 | 17287 | 333K | 2674K | 480-2160 | 1-279 | 1-10 | 1 |
| GMOT-40 | 10 | 40 | 2026 | 256K | 9K | 480-1080 | 3-24.2 | 10-128 | 24-30 |
| OV-TAO-val | 330 | 988 | 5473 | 113K | 36K | 480-2160 | 15-63 | 1-11 | 1 |
| OV-TAO-test | 357 | 1419 | 7946 | 166K | 52K | 480-2160 | 10-59 | 1-11 | 1 |
| OVT-B (Ours) | 1048 | 1973 | 13686 | 673K | 88K | 360-1440 | 1-220 | 2-86 | 5-30 |

As shown in Table 1, we summary the data statistics of the classical MOT datasets, *i.e.*, MOT17 [37], MOT20 [38], KITTI [36], DanceTrack [39] and UAVDT [48], and two new generic MOT datasets, *i.e.*,

TAO [44], GMOT-40 [43] and the existing open-vocabulary MOT dataset, *i.e.*, OV-TAO-validation [5], OV-TAO-test [5]. We can see that, the proposed OVT-B includes the most object classes. Also, the data scale of OVT-B is quite large compared to existing datasets. Note that, although TAO is larger containing more videos and tracks, its annotation frame rate is only 1 fps. So the annotated frames (which can be used for evaluation) in TAO are much fewer than in our dataset.

## 3 OVT Benchmark

To better evaluate the open-vocabulary tracking task, we establish a new large-scale dataset named OVT-B (Open-Vocabulary Tracking Benchmark). In this section, we present the video collection and annotation of OVT-B, as well as provide statistical information about this dataset and the comparisons with other datasets.

### 3.1 Data Collection

Similar to the TAO dataset, we sourced video data from existing datasets to construct the OVT-B. The selection criteria for video data were as follows:
• Each video must contain multiple objects;
• The dataset should represent a variety of categories;
• Most objects must be in motion, providing trajectory information;
• The data must be original and not derived from other datasets.

Based on these criteria, we selected seven datasets previously utilized for different tasks, including multi-object tracking (MOT), video instance segmentation (VIS), and video object detection (VOD), to create OVT-B. These datasets are AnimalTrack [49], GMOT-40 [43], LV-VIS [50], OVIS [51], UVO [52], YouTube-VIS [53], and ImageNet-VID [54]. Based on these datasets, we excluded the sequences featuring only background categories, non-specific categories, and unknown categories, retaining only those containing at least two objects. To more closely mirror various real-world scenarios and present more challenging scenes, we preserved the original resolution, duration, and annotation frame rate of the videos. It is important to note that the OVMOT task employs pre-trained OVOD models, typically without training specifically on the OVMOT dataset. This underscores the critical need for a robust evaluation benchmark. Consequently, the proposed OVT-B does not partition the training/testing dataset and serves exclusively as a comprehensive testing set.

### 3.2 Dataset Annotation

Creating unified annotations for sequences from different datasets presents several challenges, surpassing the complexity of annotating homogeneous sequences. ❶ *Annotation format difference*: Firstly, annotation formats and file storage conventions vary significantly across tasks. For instance, MOT datasets generally adhere to the MOTChallenge [37, 38] format, VIS datasets to the MS COCO [55] format, and VOD datasets to the ImageNet-VID [54] format. Additionally, even within the same task category, annotation formats may differ, necessitating bespoke processing for each dataset. ❷ *Category definition differences*: Secondly, category definitions across datasets are not uniform. Common issues include single objects corresponding to vocabularies of different granularities (*e.g.*, 'livestock' *vs.* 'pig'), objects associated with multiple vocabularies that have the same semantic meaning (*e.g.*, 'couch' and 'sofa'), and vocabularies that encompass multiple meanings, thus corresponding to different objects (*e.g.*, 'bow' as an ornament *vs.* 'bow' as a weapon). ❸ *Occlusion annotation manner*: Thirdly, the representation of completely occluded objects varies between datasets. For example, MOT datasets typically predict the motion position of an occluded object, whereas VIS datasets might label the position as null.

① To address the above challenge of annotation format differences, we initially stripped unnecessary information from the original dataset annotations specific to MOT and converted these annotations into a uniform format. Specifically, we adapted the annotations to match the TAO protocol [44], standardizing file formats and simplifying redundant data.

② For the category definition difference, we meticulously reviewed all categories in the original seven datasets, merged synonyms, applied semantic constraints to polysemous terms, and eliminated some generic categories, ultimately preserving 1,048 distinct categories.

③ To handle the annotation of occluded objects, following the TAO protocol, we uniformly set the positions of completely occluded objects as null. This way, in our dataset, objects with partial occlusion are reserved, and the objects completely occluded are not annotated, but if they reappear, they retain their original track ID.

We clarified that each above step of the annotation process was rigorously managed, involving professional manual annotation, double-checking, and correction to ensure accuracy and consistency. For a such large-scale dataset with abundant categories, the data cleaning and annotation is quite labor intensive.

### 3.3 Dataset Statistics and Comparison

Next, we show the dataset statistics of OVB-B from multiple dimensions in detail. Since OV-MOT-val and OV-MOT-test [5] have the same distribution and similar attributes, we compared one of them to OVT-B, to show the advantages of the proposed OVT-B, including diverse categories, large scale, dense annotations, and numerous targets.

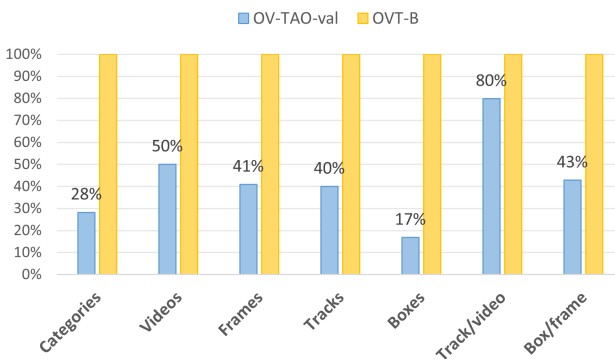

Figure 1: Comparison of OV-TAO-val and OVT-B.

• **Diverse Categories**: The OVT-B dataset contains 1,048 categories, which are divided into 534 base categories and 514 new categories that are distinct from the base ones. The base categories are derived from the frequent and common categories in the LVIS dataset. Compared to existing MOT datasets such as MOT17 [37] (1 category), KITTI [36] (5 categories), and UAVDT-MOT [48] (3 categories), OVT-B has a significantly larger number of categories, even exceeding the basic dataset of OV-TAO-val [5], *i.e.*, TAO [44] (833 categories). In the domain of open vocabulary multi-object tracking (OVMOT), TAO-val only utilizes the categories overlapping with LVIS [8], consisting of 295 base categories and 35 novel categories. However, in open vocabulary scenarios, new categories do not undergo pre-training, hence we believe there is no need to confine novel categories to the predefined novel categories of LVIS [8].

This way, in OVT-B, we set the base classes following the setting in LVIS, but the novel classes are out of the scope of that in LVIS. Note that, we still guarantee that the novel classes in OVT-B have no overlap with the base classes in LVIS, *i.e.*, the novel classes are unseen before. As shown in Figure 1, the categories in OV-TAO-val is only about 28% of that in OVT-B. Besides abundance, the category set in our dataset ensures a more balanced ratio between base and new categories, more accurately evaluating the model's ability to recognize new categories. We show the word cloud of the categories in OVT-B in Figure 2.

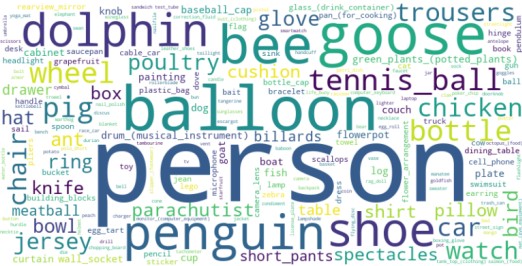

Figure 2: Word cloud of OVT-B categories.

• **Large Scale**: OVT-B features a larger number of annotated frames, trajectories, bounding boxes, and video counts, making it a dataset of a significantly larger scale, see Figure 1. This meets the current needs for evaluating rapidly increasing model sizes and capabilities.

• **Dense Targets**: As shown in Figure 3(a), the range of video length in OVT-B is much larger than that in OV-TAO-val. As shown in Figure 3(b), the maximal resolution in OVT-B is lower than OV-TAO-val. We further calculate the mean, median, and mode of the image resolution in them. The average resolution of OVT-B (724 p) is slightly lower than that of OV-TAO-val (788 p). But both the

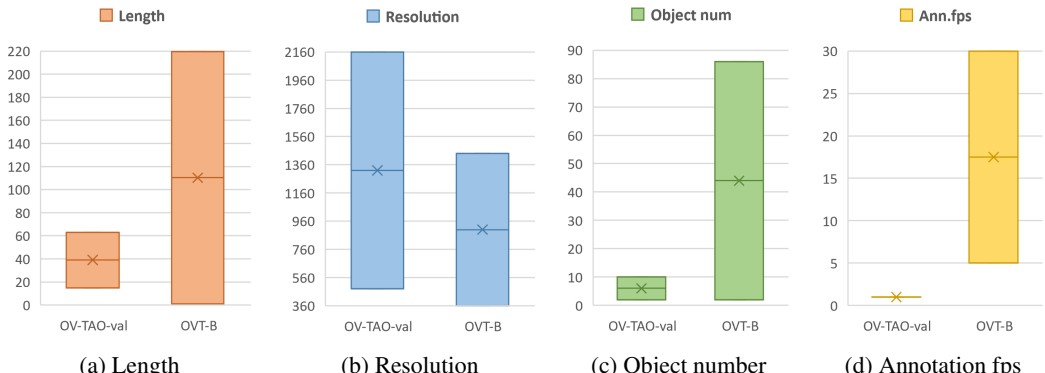

| (a) Length | (b) Resolution | (c) Object number | (d) Annotation fps |

Figure 3: Comparison of OV-TAO-val with OVT-B in scope.

median and mode of the resolution in OVT-B are the same as those of OV-TAO-val. In terms of the image resolution, OVT-B is comparable with OV-TAO-val. As in Figure 3(c), unlike the OV-TAO-val dataset, which limits up to 10 targets per frame, OVT-B does not impose such a limit, thus boasting a higher number of video targets. In OVT-B, the maximum number of targets per frame can reach 86. A higher number of targets per frame implies increased scene congestion and complexity, allowing for an evaluation of models' performance in complex environments.

• **Complete Annotations**: As shown in Figure 4, unlike OV-TAO-val, which focuses on annotating prominent targets, our dataset includes a certain number of occlusion and dense cases, providing a more diverse set of evaluation scenarios. Besides, as shown in Figure 3(d), OVT-B possesses a higher annotation frame

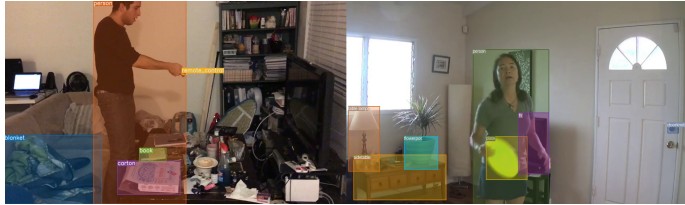

Figure 4: Screenshots of annotations of OV-TAO-val and OVT-B.

rate, with the lowest being 5 and the highest reaching 30. This enables models to fully utilize the information in every video frame for tracking and recognition. By contrast, the annotation frame rate of OV-TAO-val is 1, leaving many frames unannotated. It does not meet the need to sufficiently evaluate the performance of models.

## 3.4 Dataset Attributes

Next, we further describe the detailed attributes of the OVT-B. First, the videos in our dataset are various with many challenges in terms of the tracking task, *i.e.*, the object with out-of-view, fast motion, shape change, or different degrees of occlusion.

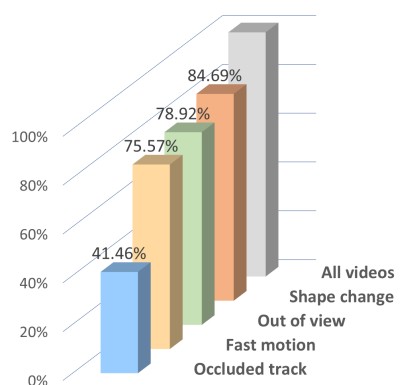

Figure 5: Ratio of videos with attributes.

Specifically, following previous works for object tracking [38], the above attributes are defined as following four aspects:

- *Occluded track* – The target is obscured or lost during part of the trajectory.
- *Fast motion* – The target moves more than 1/25 of the image width between two frames.
- *Out of view* – Part of the target is outside the image boundary.
- *Shape change* – A change of aspect ratio of the target greater than 1/5 between two frames.

As shown in Figure 5, OVT-B presents significant challenges for multi-object tracking (MOT) tasks. It re-

quires methods to possess the ability to handle occlusion, perform accurate motion prediction for fast-moving objects, and correctly classify and associate targets when they are partially missing or undergoing shape changes.

We also investigate the attributes of the objects in OVT-B. Specifically, we analyze the size and shape of the objects and the length of the tracks, as below.

• *Object size*:
- Large objects – Occupy more than 1/2 of the image area.
- Medium objects – Occupy less than 1/2 but more than 1/10 of the image area.
- Small objects – Occupy less than 1/10 of the image area.
• *Object shape*:
- Complex shapes – Have aspect ratios greater than 5 or less than 1/5.
- Intermediate shapes – Have aspect ratios less than 5 but greater than 2, or greater than 1/5 but less than 1/2.
- Normal shapes – Have aspect ratios less than 2 but greater than 1/2.
• *Track length*:
- Long tracks – The trajectory length exceeds 4/5 of the video length.
- Medium tracks – The trajectory length is less than 4/5 of the video length but more than 1/5.
- Short tracks – The trajectory length is less than 1/5 of the video length.

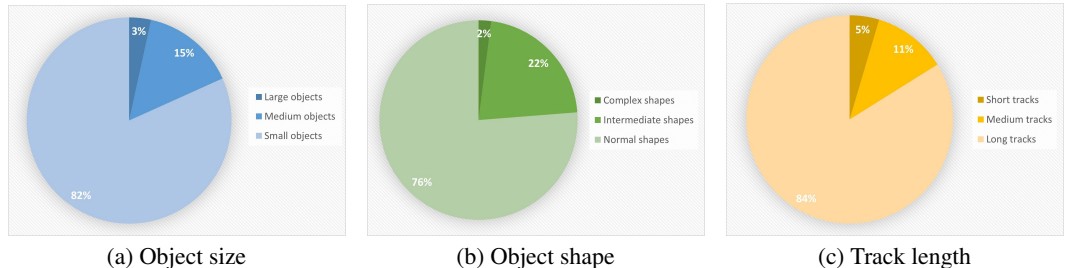

(a) Object size        (b) Object shape        (c) Track length

Figure 6: Proportion of different object sizes, object shapes, and track lengths in OVT-B.

As shown in Figure 6, we can first see that, similar to most MOT datasets, the small objects take up the major proportion in our dataset. OVT-B also includes about 20% medium/large-size objects, which is more plentiful than previous MOT datasets. *e.g.*, MOT 20. Similarly. To the object shape, the normal shapes naturally take up the majority of targets. Our dataset also contains about a quarter of objects with abnormal shapes, which can increase the richness of data. Finally, we can see that most of the trajectories in OVT-B are long, with a portion being medium or short. This can better evaluate the performance of the tracking tasks. These data attributes and distribution reflect the diversity of targets and trajectories in OVT-B, as well as a comprehensive range of tracking scenarios.

### 3.5 Metrics

We use the tracking-every-thing accuracy (TETA) [6] as the evaluation metric, which is calculated from three independent scores. First, localization accuracy (LocA) is calculated based on the matching of annotation boxes and predicted boxes, $LocA = \frac{|TPL|}{|TPL|+|FPL|+|FNL|}$, followed by the calculation of classification accuracy (ClsA) based on TPL with good localization results, $ClsA = \frac{|TPC|}{|TPC|+|FPC|+|FNC|}$, Subsequently, association accuracy (AssA) is calculated based on TPL with good localization results, $AssA = \frac{1}{|\mathrm{TPL}|}\sum_{b\in\mathrm{TPL}}\frac{|\mathrm{TPA}(b)|}{|\mathrm{TPA}(b)|+|\mathrm{FPA}(b)|+|\mathrm{FNA}(b)|}$, Ultimately, TETA is obtained by taking the arithmetic mean of the three accuracies, $TETA = \frac{LocA+ClsA+AssA}{3}$. In OVMOT, following the evaluation method of OVOD, TETA is calculated separately for base and novel classes.

# 4 OVTrack+: A New Baseline

OVTrack [5], as the first and alone public tracker for OVMOT, uses only the appearance feature for the association. In this section, we introduced a simple yet effective baseline incorporating a motion model into open-vocabulary multi-object tracking, using motion information and appearance features as cues for association.

**Integrating motion model for OVTrack.** In addressing the challenge of open-vocabulary multi-object tracking, we believe that the integration of a target motion model is advantageous for association tasks due to its category-agnostic nature. While the proliferation of categories introduces specific motion patterns that may challenge the assumptions inherent to the classical motion model, *e.g.*, the Kalman filter, this model nonetheless offers valuable supervisory data that aids in the association process for most categories. Experimental evidence, however, indicates that reliance solely on the motion model for object tracking is suboptimal. Drawing on these insights, we developed a method namely OVTrack+ that eliminates the decision threshold and integrates appearance features with motion information.

Before distance computation, the IoU distance between every pair of detected objects $r \in R$ is calculated. Objects with an IoU score greater than the threshold are considered that occlusion and the one with the lower confidence is removed. For each track $\tau \in T$, we first use a Kalman filter to predict the motion position, resulting in $p_\tau$, and then calculate the IoU distance between $p_\tau$ and the remaining detected objects $p_r$. Next, we compute the appearance distance between the stored appearance embeddings $q_\tau$ in the track and the detected object's embeddings $q_r$. The appearance distance $D_{app}$ is calculated using a weighted combination of the bi-softmax score $S_{bi}$[56] and the cosine score $S_{cos}$

$$S_{\text{bi}}(\tau, r) = \frac{1}{2} \left[ \frac{\exp(q_r \cdot q_\tau)}{\sum_{r' \in R} \exp(q_{r'} \cdot q_\tau)} + \frac{\exp(q_r \cdot q_\tau)}{\sum_{\tau' \in \mathcal{T}} \exp(q_r \cdot q_{\tau'})} \right], \tag{1}$$

$$S_{\text{cos}}(\tau, r) = \frac{\tau}{\|\tau\|_2} \cdot \left(\frac{r}{\|r\|_2}\right)^\top, D_{\text{app}} = \left( \frac{1}{2} (1 + S_{\text{cos}}) + S_{\text{bi}} \right). \tag{2}$$

The final distance matrix $D$ is obtained by weighting the IoU distance $D_{IoU}$ and the appearance distance $D_{app}$

$$D = D_{\text{app}} \cdot (1 - w) + w \cdot D_{\text{IoU}}. \tag{3}$$

Finally, the Hungarian algorithm is applied to the distance matrix for optimal matching. Matches with distances below the threshold are considered successful, while those above the threshold are assigned new IDs. Note that no detected objects are discarded in the open vocabulary tracker due to the generally low and unreliable classification confidence. For the tracks successfully matched, the appearance embeddings $e^{k-1}$ are updated by the Exponential Moving Average (EMA) mechanism

$$e_i^k = \alpha e_i^{k-1} + (1 - \alpha) f_i^k. \tag{4}$$

**Implementation details.** In our approach, we utilize a two-stage detector that is identical to our baseline method OVTrack [5], employing ResNet50 [57] coupled with a Feature Pyramid Network (FPN) [58], the detection head from ViLD [3] and the tracking head from OVTrack. During inference, the training weights from OVTrack are used along with text embeddings generated by Detpro [32]. We maintain the same operational settings, setting the match distance threshold at 0.5, and the IoU threshold for track initialization at 0.3. To enhance tracking performance, our newly integrated motion model incorporates a memory frame count of 30 frames, a momentum parameter $\alpha$ in Eq. (4) for update embeddings of 0.2, and a motion distance weight $w$ in Eq. (3) of 0.03.

# 5 Experiments

## 5.1 Comparison with State-of-The-Art Methods

● *ByteTrack[59]*: It is a renowned motion-based model that relies solely on high-performance detectors and motion information, achieving high running speed and state-of-the-art performance. It utilizes low-score detection boxes by initially matching high-confidence detections, followed by an

association with the low-confidence detections.

• *OC-SORT[60]*: It is derived from ByteTrack[59], also a motion-based model, and achieves new state-of-the-art performance after ByteTrack. It enhances tracking robustness in non-linear motion scenarios and mitigates the impact of object occlusion or disappearance by relying heavily on detections.

• *StrongSORT[14]*: It is a hybrid model combining motion and appearance features, by equipping DeepSORT[2] with advanced components. It introduces a simple yet effective baseline and attains state-of-the-art performance when proposed.

• *OVTrack[5]*: It is derived from QDTrack[56], which is a pure appearance-based model, and also a SOTA model known for its simplicity and effectiveness, without bells and whistles.

Table 2: Open-vocabulary MOT comparison results on OVT-B.

| Method | All | | | | Base | | | | Novel | | | |
|---|---|---|---|---|---|---|---|---|---|---|---|---|
| | TETA | LocA | AssA | ClsA | TETA | LocA | AssA | ClsA | TETA | LocA | AssA | ClsA |
| ByteTrack [59] | 20.1 | 36.1 | 12.4 | 11.9 | 20.6 | 35.6 | 12.7 | 13.4 | 19.6 | 36.6 | 12.0 | **10.3** |
| OC-SORT  [60] | 16.0 | 31.2 | 4.3 | **12.3** | 16.5 | 31.0 | 4.4 | **14.3** | 15.4 | 31.4 | 4.3 | **10.3** |
| StrongSORT [14] | 24.8 | 31.6 | 30.7 | 12.2 | 25.7 | 31.4 | 31.6 | 14.2 | 23.9 | 31.8 | 29.7 | **10.3** |
| OVTrack [5] | 46.1 | 60.8 | 66.1 | 11.5 | 46.8 | 60.5 | 66.7 | 13.4 | 45.5 | 61.1 | 65.5 | 9.6 |
| OVTrack+ | **47.0** | **62.0** | **67.7** | 11.3 | **47.6** | **61.6** | **68.2** | 13.2 | **46.4** | **62.5** | **67.3** | 9.4 |

Table 3: Open-vocabulary MOT comparison results on OV-TAO-val.

| Method | All | | | | Base | | | | Novel | | | |
|---|---|---|---|---|---|---|---|---|---|---|---|---|
| | TETA | LocA | AssA | ClsA | TETA | LocA | AssA | ClsA | TETA | LocA | AssA | ClsA |
| ByteTrack [59] | 20.1 | 36.9 | 6.0 | **17.6** | 20.9 | 37.0 | 5.9 | **19.7** | 14.7 | 36.0 | 6.1 | 1.8 |
| OC-SORT  [60] | 24.3 | 52.1 | 6.0 | 14.8 | 25.1 | 52.7 | 6.1 | 16.5 | 18.5 | 48.1 | 5.4 | **2.1** |
| StrongSORT [14] | 23.4 | 41.6 | 13.5 | 15.2 | 24.4 | 42.3 | 13.7 | 17.0 | 16.6 | 36.4 | 11.6 | 1.7 |
| OVTrack [5] | 36.1 | 53.8 | 37.3 | 17.3 | 37.1 | 54.2 | 37.8 | 19.4 | 28.8 | 51.2 | 33.7 | 1.5 |
| OVTrack+ | **38.4** | **57.5** | **40.8** | 16.9 | **39.2** | **57.5** | **41.0** | 18.9 | **32.5** | **57.0** | **38.7** | 1.8 |

We present the MOT evaluation results of open vocabulary multi-object tracking on the OV-TAO-val and OVT-B, see Table 2 and Table 3. Compared to the OVTrack, OVTrack+ achieves higher performance on TETA, LocA, and AssA. In terms of ClsA, OVTrack+ experiences a slight decline in performance, indicating that the inclusion of the motion model does not contribute to an improvement in classification performance.

## 5.2 In-depth Experimental Analysis

In OVT-B, all methods exhibit significantly higher AssA compared to OV-TAO-val. This indicates that the high annotation frame rate of OVT-B, by providing more densely evaluated frames, allows for a more comprehensive and detailed assessment, thereby reducing cumulative error scores. Consequently, it reflects the actual accuracy of the model in target association tasks. Moreover, the performance of various methods is more consistent in OVT-B, suggesting that the scenes and object characteristics in OVT-B are more uniform. This uniformity reduces the variability in algorithm performance under different conditions, facilitating more accurate evaluation and comparison of different methods. Additionally, in OV-TAO-val, OVTrack+ significantly improved performance by incorporating a motion model in association, demonstrating the potential of motion feature-based methods in this dataset. On the other hand, in OVT-B, methods utilizing appearance features consistently achieve higher AssA than those relying solely on motion features. This indicates that objects in OVT-B have more distinctive appearance characteristics. Therefore, OVT-B can effectively complement OV-TAO-val by providing a more comprehensive evaluation.

## 5.3 Discussion

**New challenges and lessons.** In this discussion, we delineate the distinctions between open-vocabulary multi-object tracking (OVMOT) and traditional MOT, and what we learned from practice. The introduction of a lot of categories and the low performance of OVD models significantly diminish the reliability of category confidence, due to the closely similar and low confidence scores across

categories. It impairs the effectiveness of using classification confidence as a metric during the association phase – a common practice in traditional MOT methods. Notably, training solely on the categories from the training set still introduces a bias between the appearance quality of base and novel classes, which does not manifest when employing only motion models for association.

Moreover, the use of category information as a cue to supervise open vocabulary association has proven impractical. The impracticality arises from two primary factors: the insufficient reliability of category information provided by detection models, and the minimal distinctions between categories, coupled with the less pronounced than intra-category variances. Consequently, these observations highlight the imperative need to further develop and refine the association mechanisms tailored for the OVMOT context, where conventional strategies falter due to the unique complexities introduced by open vocabulary settings.

**Current situation and future outlook.** Based on the results of the current state of the object tracking methods on the proposed dataset, we have some thoughts in the following.

In terms of the problem, as shown in Table 2, we can clearly see that the object classification results, especially for the novel class, are very low. This demonstrates that this is a very challenging problem having lots of room for improvement. So, how to improve the novel-class object classification ability of the open-set object tracking method, is a difficult problem worthy of further research. Also, we find that the current methods for open-set object tracking can not handle the performance balance among different sub-tasks. We think that the three sub-tasks, i.e., localization, classification, and tracking could be complementary to each other. For example, on the one hand, the correct tracking results can help the classification task, i.e., the predicted object category should be consistent along a trajectory. On the other hand, the object classification results can also help the tracking, where the object category can be used as a cue for temporal object association during tracking.

In terms of the methods, in classical multi-object tracking, the method can be divided into two categories, i.e., tracking-by-detection methods, and joint embedding of tracking and detection based methods. Classical MOT does not require classification during detection. In the open-vocabulary setting, the detection task is more challenging requiring open-class recognition ability. This way, from our point of view, the tracking-by-detection methods would be the mainstream framework in the near future. This is because the joint feature embedding for three different tasks is very challenging. As discussed in the above (second point), we think that using the results from different tasks to complement each other may be a better solution at present. We also hope to see the first effective joint embedding based method for OVMOT.

Besides, in terms of the evaluation metric, the existing overall metric TETA directly calculates the average of the localization, classification, and tracking accuracies. Considering the difficulty imbalance among different sub-tasks, a new metric for more reasonable evaluation may be required.

Finally, a more recent work [61] aims to handle the object classification task in OVTrack as the recognition problem and proposes a new task namely open-corpus tracking (OCTrack), which may be a further step of OVTrack.

# 6 Conclusion

In this work, we have built a new large-scale benchmark – OVT-B for the emerging open-vocabulary multi-object tracking (OVMOT). The proposed OVT-B is much larger than the only existing open-vocabulary tracking dataset OV-TAO-val dataset, regardless of video amount or category amount. The proposed OVT-B is very promising to serve as a new benchmark for the study of OVMOT. We develop a simple yet effective baseline for OVMOT that integrates the motion features for object tracking. Experimental results have verified the usefulness of the proposed OVT-B. We have also delineated the distinctions between OVMOT and traditional MOT and provided some experiences and lessons to tackle the new challenges of OVMOT, as well as some outlook in the future. Through the above effort, we aim to pave the way for further research on this topic.

# Acknowledgment

This work was supported in part by the National Natural Science Foundation of China (NSFC) under Grant 62402490 and the China Postdoctoral Science Foundation 2024M753397.

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
