# OVT-B: A New Large-Scale Benchmark for Open-Vocabulary Multi-Object Tracking
## *Supplementary Material*

**Haiji Liang**[1], **Ruize Han**[2,3*]
[1]School of Software Technology, Zhejiang University
[2]Shenzhen Institute of Advanced Technology, Chinese Academy of Sciences
[3]Department of Computer Science, City University of Hong Kong
coolsea@zju.edu.cn, rz.han@siat.ac.cn

## 1 Datasheets for OVT-B dataset

### Motivation

**For what purpose was the dataset created?** Was there a specific task in mind? Was there a specific gap that needed to be filled? Please provide a description.

In the current task of open-vocabulary multi-object tracking (OVMOT), there is only one benchmark available, which lacks high-quality, large-scale datasets. The existing dataset suffers from several limitations, including insufficient categories, limited video data, and a significant imbalance between base classes and novel classes. These deficiencies make it inadequate for supporting the evaluation of new OVMOT models. Our proposed dataset aims to provide a more comprehensive evaluation platform for the OVMOT task.

**Who created this dataset (e.g., which team, research group) and on behalf of which entity (e.g., company, institution, organization)?**

This dataset was constructed by collecting and extracting data from seven other datasets and applying unified annotations. This work was completed by Haiji Liang and Ruize Han.

**Who funded the creation of the dataset?** If there is an associated grant, please provide the name of the grantor and the grant name and number.

No funding was received for the creation of this dataset.

**Any other comments?**

No additional comments.

### Composition

**What do the instances that comprise the dataset represent (e.g., documents, photos, people, countries)?** Are there multiple types of instances (e.g., movies, users, and ratings; people and interactions between them; nodes and edges)? Please provide a description.

---

[*]Corresponding Author.

38th Conference on Neural Information Processing Systems (NeurIPS 2024) Track on Datasets and Benchmarks.

The instances that comprise the dataset are video frames along with their respective annotation information. Each instance represents an individual frame from a video, accompanied by detailed annotations that describe the objects present in that frame, their positions, and other relevant attributes. The dataset includes two types of instances, such as:

- Video Frames: Each frame includes multiple objects from the videos.

- Annotations: Metadata associated with each frame, including bounding boxes, object labels, and other relevant attributes.

### How many instances are there in total (of each type, if appropriate)?

The dataset consists of a total of 88K video frames and 673K bounding box annotations.

### Does the dataset contain all possible instances or is it a sample (not necessarily random) of instances from a larger set? If the dataset is a sample, then what is the larger set? Is the sample representative of the larger set (e.g., geographic coverage)? If so, please describe how this representativeness was validated/verified. If it is not representative of the larger set, please describe why not (e.g., to cover a more diverse range of instances, because instances were withheld or unavailable).

The dataset is a sample of instances from a larger set. The larger set consists of all available video frames and annotations from the original seven datasets used to construct this dataset. Based on task requirements, we excluded videos with only one target and those with non-representative categories. The selected videos exhibit a diverse range of object categories, complex tracking scenarios, and unique object attributes, including a varied distribution of video lengths.

Sampling Strategy: A specific sampling strategy was implemented to ensure the dataset's representativeness:

Initial Selection (S1): For categories with less than 10 occurrences in the total video set S0, all instances of those categories were selected, forming the initial video set S1. Recount and Additional Selection (S2): After the initial selection, the occurrence of each category within S1 was recounted. For categories still under 10 occurrences, additional videos from the remaining set (S0 - S1) were selected until each category had at least 10 occurrences, forming the updated video set S2. The new video set thus became S1 + S2. Iterative Process: This process was repeated, each time ensuring that categories with fewer than 10 occurrences were supplemented from the remaining unselected videos until all categories either had fewer than 10 total occurrences (and thus all instances were selected) or had at least 10 occurrences in the final dataset. Representativeness of the Sample: The sample is representative of the larger set in terms of the variety of object categories, scene types, and motion patterns present in the videos. This representativeness was validated by:

Distribution Check: After applying the sampling strategy, we re-evaluated the distribution of categories and found that it still reflects common distributions. For example, categories with high occurrences in S0, such as "person," maintained high distributions in the sampled dataset, indicating the sample's representativeness. Coverage of Diverse Instances: The selected videos cover a wide range of object categories, complex tracking scenarios, and unique object attributes, ensuring the dataset's comprehensiveness and suitability for diverse analysis and evaluation needs. In conclusion, the dataset, while being a sample, is designed to be highly representative of the larger set. This careful selection process ensures that it can effectively support comprehensive evaluation and research in open-vocabulary multi-object tracking (OVMOT).

### What data does each instance consist of? "Raw" data (e.g., unprocessed text or images) or features? In either case, please provide a description.

The data for each instance in our dataset consists of "Raw" video frames.

### Is there a label or target associated with each instance? If so, please provide a description.

The video frames within a single video collectively describe the movement of several objects.

**Is any information missing from individual instances?** If so, please provide a description, explaining why this information is missing (e.g., because it was unavailable). This does not include intentionally removed information, but might include, e.g., redacted text.

No information is missing from individual instances.

**Are relationships between individual instances made explicit (e.g., users' movie ratings, social network links)?** If so, please describe how these relationships are made explicit.

The relationships between individual instances are made explicit by the fact that they all belong to the same video. Each set of video frames collectively describes the movement of objects within that particular video. There are no explicit relationships between instances from different videos.

**Are there recommended data splits (e.g., training, development/validation, testing)?** If so, please provide a description of these splits, explaining the rationale behind them.

Our data is solely used for validation. No training set has been created.

**Are there any errors, sources of noise, or redundancies in the dataset?** If so, please provide a description.

There are no errors, sources of noise, or redundancies in the dataset.

**Is the dataset self-contained, or does it link to or otherwise rely on external resources (e.g., websites, tweets, other datasets)?** If it links to or relies on external resources, a) are there guarantees that they will exist, and remain constant, over time; b) are there official archival versions of the complete dataset (i.e., including the external resources as they existed at the time the dataset was created); c) are there any restrictions (e.g., licenses, fees) associated with any of the external resources that might apply to a future user? Please provide descriptions of all external resources and any restrictions associated with them, as well as links or other access points, as appropriate.

The dataset is self-contained and does not rely on external resources.

**Does the dataset contain data that might be considered confidential (e.g., data that is protected by legal privilege or by doctor-patient confidentiality, data that includes the content of individuals non-public communications)?** If so, please provide a description.

The dataset does not contain any data that might be considered confidential.

**Does the dataset contain data that, if viewed directly, might be offensive, insulting, threatening, or might otherwise cause anxiety?** If so, please describe why.

The dataset does not contain any data that, if viewed directly, might be offensive, insulting, threatening, or might otherwise cause anxiety.

**Does the dataset relate to people?** If not, you may skip the remaining questions in this section.

Yes, the dataset relates to people as it includes objects categorized as "person".

**Does the dataset identify any subpopulations (e.g., by age, gender)?** If so, please describe how these subpopulations are identified and provide a description of their respective distributions within the dataset.

No, the dataset does not identify any subpopulations (e.g., by age, gender).

**Is it possible to identify individuals (i.e., one or more natural persons), either directly or indirectly (i.e., in combination with other data) from the dataset?** If so, please describe how.

Yes, it is possible to identify individuals from the dataset. While the dataset predominantly features video footage that does not focus on faces, there are some instances where videos may contain

close-up shots of individuals' faces. These instances, though infrequent, could potentially be used to identify individuals when combined with other data.

**Does the dataset contain data that might be considered sensitive in any way (e.g., data that reveals racial or ethnic origins, sexual orientations, religious beliefs, political opinions or union memberships, or locations; financial or health data; biometric or genetic data; forms of government identification, such as social security numbers; criminal history)?** If so, please provide a description.

No, the dataset does not contain any data that might be considered sensitive in any way (e.g., data that reveals racial or ethnic origins, sexual orientations, religious beliefs, political opinions or union memberships, locations; financial or health data; biometric or genetic data; forms of government identification, such as social security numbers; criminal history).

**Any other comments?**

No additional comments.

---

**Collection Process**

**How was the data associated with each instance acquired?** Was the data directly observable (e.g., raw text, movie ratings), reported by subjects (e.g., survey responses), or indirectly inferred/derived from other data (e.g., part-of-speech tags, model-based guesses for age or language)? If data was reported by subjects or indirectly inferred/derived from other data, was the data validated/verified? If so, please describe how.

The videos and video frames in the dataset were directly acquired from the seven original datasets.

**What mechanisms or procedures were used to collect the data (e.g., hardware apparatus or sensor, manual human curation, software program, software API)?** How were these mechanisms or procedures validated?

The data was collected by writing code to programmatically filter and extract relevant videos and video frames from the original datasets based on predefined criteria. These mechanisms were validated by ensuring that the filtering criteria accurately reflected the requirements of the dataset and by manually reviewing a subset of the extracted data to confirm its correctness.

**If the dataset is a sample from a larger set, what was the sampling strategy (e.g., deterministic, probabilistic with specific sampling probabilities)?**

The dataset was sampled based on several OVMOT benchmark criteria. Specifically, the selection criteria included: the number of targets must be greater than 1, the number of frames must be greater than 5, the resolution must be higher than 360p, and the object categories must not belong to a major or generic category, 'unknown' category, or 'background' category.

Additionally, a specific sampling strategy was implemented based on the frequency of each category's occurrence in the videos. The strategy is as follows:

Initial Selection (S1): For categories with less than 10 occurrences in the total video set S0, all instances of those categories were selected, forming the initial video set S1. Recount and Additional Selection (S2): After the initial selection, the occurrence of each category within S1 was recounted. For categories still under 10 occurrences, additional videos from the remaining set (S0 - S1) were selected until each category had at least 10 occurrences, forming the updated video set S2. The new video set thus became S1 + S2. Iterative Process: This process was repeated, each time ensuring that categories with fewer than 10 occurrences were supplemented from the remaining unselected videos until all categories either had fewer than 10 total occurrences (and thus all instances were selected) or had at least 10 occurrences in the final dataset. This iterative approach ensures that the dataset is comprehensive and representative, with sufficient examples for each category, while adhering to the initial selection criteria.

**Who was involved in the data collection process (e.g., students, crowdworkers, contractors) and how were they compensated (e.g., how much were crowdworkers paid)?**

The data collection process was carried out by a student. No compensation was provided.

**Over what timeframe was the data collected? Does this timeframe match the creation timeframe of the data associated with the instances (e.g., recent crawl of old news articles)?** If not, please describe the timeframe in which the data associated with the instances was created.

The data was collected over a timeframe from [2023.10] to [2024.1]. This timeframe does not match the creation timeframe of the original data associated with the instances, as the data was extracted from seven existing datasets. The original data in these datasets was created at various times prior to this collection period.

**Were any ethical review processes conducted (e.g., by an institutional review board)?** If so, please provide a description of these review processes, including the outcomes, as well as a link or other access point to any supporting documentation.

No, the dataset did not undergo an ethical review process by an Institutional Review Board (IRB). The data was collected from publicly available datasets and did not involve any direct interaction with human subjects or sensitive personal information.

**Does the dataset relate to people?** If not, you may skip the remaining questions in this section.

Yes, the dataset relates to people as it includes objects categorized as "person".

**Did you collect the data from the individuals in question directly, or obtain it via third parties or other sources (e.g., websites)?**

We obtained the original data from the websites of the seven datasets.

**Were the individuals in question notified about the data collection?** If so, please describe (or show with screenshots or other information) how notice was provided, and provide a link or other access point to, or otherwise reproduce, the exact language of the notification itself.

No, the individuals in question were not notified about the data collection because we collected the data from open-source datasets.

**Did the individuals in question consent to the collection and use of their data?** If so, please describe (or show with screenshots or other information) how consent was requested and provided, and provide a link or other access point to, or otherwise reproduce, the exact language to which the individuals consented.

The individuals in question did not provide direct consent to us for the collection and use of their data because we collected the data from open-source datasets. These datasets are publicly available and have been shared under licenses that permit their use. The consent for data collection and use was managed by the original dataset providers as part of their data collection process. For specific details on how consent was obtained, please refer to the documentation provided by the original datasets.

**If consent was obtained, were the consenting individuals provided with a mechanism to revoke their consent in the future or for certain uses?** If so, please provide a description, as well as a link or other access point to the mechanism (if appropriate).

Since we collected the data from open-source datasets, any mechanisms for revoking consent would be managed by the original dataset providers. These datasets are shared under licenses that outline the terms of use and any rights of the individuals to revoke their consent. For specific details on how consent revocation is handled, please refer to the documentation provided by the original datasets.

**Has an analysis of the potential impact of the dataset and its use on data subjects (e.g., a data protection impact analysis) been conducted?** If so, please provide a description of this analysis, including the outcomes, as well as a link or other access point to any supporting documentation.

No, an analysis of the potential impact of the dataset and its use on data subjects (e.g., a data protection impact analysis) has not been conducted. This is because the dataset was collected from open-source datasets that are publicly available and shared under licenses that permit their use. As the data was already publicly accessible and shared under open-source licenses, the original dataset providers would have been responsible for any necessary impact analyses.

For more information on the data protection and impact assessments conducted by the original dataset providers, please refer to the documentation provided by the original datasets.

**Any other comments?**

No additional comments.

---



**Preprocessing/cleaning/labeling**



**Was any preprocessing/cleaning/labeling of the data done (e.g., discretization or bucketing, tokenization, part-of-speech tagging, SIFT feature extraction, removal of instances, processing of missing values)?** If so, please provide a description. If not, you may skip the remainder of the questions in this section.

Yes, preprocessing, cleaning, and labeling of the data were performed. Specifically, we removed certain instances and standardized the annotation format and categories.

**Was the "raw" data saved in addition to the preprocessed/cleaned/labeled data (e.g., to support unanticipated future uses)?** If so, please provide a link or other access point to the "raw" data.

Yes, the raw data is available. I can provide the download links to the seven original datasets from which the raw data was collected. These links can be used as access points to the raw data.

1. AnimalTrack: https://hengfan2010.github.io/projects/AnimalTrack
2. GMOT-40: https://github.com/Spritea/GMOT40
3. ImageNet-VID: https://www.image-net.org/challenges/LSVRC/
4. LV-VIS: https://github.com/haochenheheda/LVVIS
5. OVIS: http://songbai.site/ovis
6. UVO: https://sites.google.com/view/unidentified-video-object
7. YouTube-VIS: https://youtube-vos.org/dataset/vis/

**Is the software used to preprocess/clean/label the instances available?** If so, please provide a link or other access point.

No, there isn't available software. The preprocessing and labeling were performed using custom scripts that I wrote.

**Any other comments?**

No additional comments.

---



**Uses**



**Has the dataset been used for any tasks already?** If so, please provide a description.

Yes, this dataset has been used as a benchmark for Open-Vocabulary Multi-Object Tracking (OV-MOT) tasks.

**Is there a repository that links to any or all papers or systems that use the dataset?** If so, please provide a link or other access point.

Yes, there is a repository that links to papers and systems using the dataset. Here is the GitHub homepage for my dataset: https://github.com/Coo1Sea/OVT-B-Dataset

**What (other) tasks could the dataset be used for?**

The dataset can be used for video object recognition.

**Is there anything about the composition of the dataset or the way it was collected and preprocessed/cleaned/labeled that might impact future uses?** For example, is there anything that a future user might need to know to avoid uses that could result in unfair treatment of individuals or groups (e.g., stereotyping, quality of service issues) or other undesirable harms (e.g., financial harms, legal risks) If so, please provide a description. Is there anything a future user could do to mitigate these undesirable harms?

No, there is nothing about the composition of the dataset or the way it was collected and preprocessed/cleaned/labeled that might impact future uses.

**Are there tasks for which the dataset should not be used?** If so, please provide a description.

There are no specific tasks for which the dataset is not suitable.

**Any other comments?**

No additional comments.

---

## Distribution

**Will the dataset be distributed to third parties outside of the entity (e.g., company, institution, organization) on behalf of which the dataset was created?** If so, please provide a description.

Yes, the dataset will be distributed to third parties outside of the entity on behalf of which the dataset was created. The dataset is open-source and is intended to be used as a benchmark for the community.

**How will the dataset will be distributed (e.g., tarball on website, API, GitHub)** Does the dataset have a digital object identifier (DOI)?

The dataset will be publicly available for download on its GitHub homepage. Additionally, the dataset will be published along with our paper, and if the paper is accepted, the dataset will be assigned a Digital Object Identifier (DOI).

**When will the dataset be distributed?**

The dataset will be distributed once the paper is accepted for publication. This ensures that the dataset has been peer-reviewed and validated. Upon acceptance, the dataset will be made available on its GitHub homepage.

**Will the dataset be distributed under a copyright or other intellectual property (IP) license, and/or under applicable terms of use (ToU)?** If so, please describe this license and/or ToU, and provide a link or other access point to, or otherwise reproduce, any relevant licensing terms or ToU, as well as any fees associated with these restrictions.

Yes, the dataset will be distributed under an open-source license. Specifically, it will be released under the Apache License 2.0. This license allows users to freely use, modify, and distribute the

dataset, provided they adhere to the terms of the license. For more information on the license and its terms, please refer to the following link: https://www.apache.org/licenses/LICENSE-2.0. There are no fees associated with these restrictions.

**Have any third parties imposed IP-based or other restrictions on the data associated with the instances?** If so, please describe these restrictions, and provide a link or other access point to, or otherwise reproduce, any relevant licensing terms, as well as any fees associated with these restrictions.

No, there are no third-party IP-based or other restrictions imposed on the data associated with the instances. The original datasets are publicly available and shared under licenses that permit their use. For more details on the original datasets' licensing terms, please refer to their respective documentation.

**Do any export controls or other regulatory restrictions apply to the dataset or to individual instances?** If so, please describe these restrictions, and provide a link or other access point to, or otherwise reproduce, any supporting documentation.

No, there are no export controls or other regulatory restrictions that apply to the dataset or to individual instances. The dataset is composed of data from publicly available open-source datasets, which are shared under licenses that do not impose any such restrictions. For more details on the original datasets' licensing terms and any related documentation, please refer to their respective websites.

**Any other comments?**

No additional comments.

---

| **Maintenance** |
|:---:|

**Who will be supporting/hosting/maintaining the dataset?**

The dataset will be hosted on its GitHub homepage, and it will be maintained by the authors of the paper, Haiji Liang and Ruize Han.

**How can the owner/curator/manager of the dataset be contacted (e.g., email address)?**

Users can contact the curator of the dataset via email. The author, Haiji Liang, can be reached at: coolsea@zju.edu.cn.

**Is there an erratum?** If so, please provide a link or other access point.

There is no erratum for the dataset at this time.

**Will the dataset be updated (e.g., to correct labeling errors, add new instances, delete instances)?** If so, please describe how often, by whom, and how updates will be communicated to users (e.g., mailing list, GitHub)?

Currently, there are no plans for updates. If updates are necessary, the dataset maintainer will update the dataset on GitHub. Users will be notified of any changes through the GitHub repository.

**If the dataset relates to people, are there applicable limits on the retention of the data associated with the instances (e.g., were individuals in question told that their data would be retained for a fixed period of time and then deleted)?** If so, please describe these limits and explain how they will be enforced.

We will closely monitor the source datasets for any applicable limits on the retention of data related to people. If the original datasets impose restrictions or take actions regarding the retention of individuals' data that appear in both their datasets and ours, we will follow their lead and implement similar actions to ensure compliance.

**Will older versions of the dataset continue to be supported/hosted/maintained?** If so, please describe how. If not, please describe how its obsolescence will be communicated to users.

Currently, there is only one version of the dataset, and there are no plans for updates. If updates are made in the future, the decision to support, host, and maintain older versions of the dataset will be determined based on the specific circumstances at that time. Users will be informed about the status of older versions and any changes through updates on the GitHub repository.

**If others want to extend/augment/build on/contribute to the dataset, is there a mechanism for them to do so?** If so, please provide a description. Will these contributions be validated/verified? If so, please describe how. If not, why not? Is there a process for communicating/distributing these contributions to other users? If so, please provide a description.

We welcome contributions from others to extend, augment, or build on this dataset. Contributions can be made through GitHub issues or pull requests, or via email. It is important to note that frequent modifications are not suitable for a benchmark dataset. Therefore, we will accumulate significant updates or contributions and make consolidated updates to the dataset. Contributions will be reviewed and validated by the maintainers before being incorporated.

**Any other comments?**

No additional comments.

## 2 Appendix

### 2.1 Implementation Details

All experiments were conducted on 2 NVIDIA TITAN RTX GPUs.

The runtime for different algorithms is shown below.

Table 1: Runtime for Different Algorithms

| Tracker | ByteTrack | OC-SORT | StrongSORT | OVTrack | OVTrack+ |
|---------|-----------|---------|------------|---------|----------|
| Runtime (FPS) | 11.2 | 9.8 | 6.2 | 7.1 | 5.8 |

We selected hyperparameters using the grid search method. For the hyperparameters of OVTrack+ mentioned in our paper, our search ranges were as follows: For the *max_per_img* parameter in RCNN, we searched from 50 to 200 with an interval of 50; for the IoU threshold for track initialization, we searched from 0.2 to 0.5 with an interval of 0.1; for the match distance threshold, we searched from 0.3 to 0.6 with an interval of 0.1; for the momentum parameter $\alpha$ to update embeddings, we searched from 1 to 0.1 with an interval of 0.1; for the motion distance weight $w$, we searched from 0.01 to 0.04 with an interval of 0.01.

For other comparison methods, we only adjusted some of the threshold parameters to adapt to the new open-vocabulary scenario while retaining most of the original settings.

### 2.2 Label Format for OVT-B

To facilitate evaluation using TETA, we have adjusted the annotation format of OVT-B to be consistent with the TAO annotation format, specifically the COCO format. This ensures compatibility with multiple libraries and other evaluation algorithms.

Annotation file format:

```
{
    "images" : [image],
    "videos": [video],
    "tracks": [track],
```

```
    "annotations" : [annotation],
    "categories": [category],
}
image: {
    "id": int,
    "video_id": int,
    "file_name": str,
    "width": int,
    "height": int,
    "frame_index": int,
    "frame_id": int
}
video: {
    "id": int,
    "name": str,
    "width": int,
    "height": int,
    "neg_category_ids": [],
    "not_exhaustive_category_ids": []
}
track: {
    "id": int,
    "category_id": int,
    "video_id": int
}
category: {
    "id": int,
    "name": str,
    "synset": "unknown",
    "frequency": "r" or "b",
}
annotation: {
    "id": int,
    "image_id": int,
    "video_id": int,
    "track_id": int,
    "bbox": [x,y,width,height],
    "area": float,
    "category_id": int
}
```