# OpenReview forum: "OVT-B: A New Large-Scale Benchmark for Open-Vocabulary Multi-Object Tracking"
_NeurIPS.cc/2024/Datasets_and_Benchmarks_Track — NeurIPS 2024 Track Datasets and Benchmarks Poster_

### Official Review · Reviewer_tr32 · 2024-07-25
**OVT-B is the first large-scale open-vocabulary multi-object tracking (OVMOT) benchmark.**

**Rating:** 6
**Confidence:** 4
**Correctness:** Technically sound

**Review:**

The proposed OVT-B benchmark significantly advances the field of open-vocabulary multi-object tracking by providing a comprehensive and challenging dataset.

Pros:
1) Comprehensive and Diverse Dataset
2) Detailed Annotations: high annotation frame rate (5-30 fps)
3) Challenging Scenarios: complex scenarios including occlusions, fast motion, and shape changes.

Cons:
1) Computational Demands: it requires significant computational resources for training and evaluation
2) Low Resolution: The average resolution is lower than that of the existing OV-TAO-val dataset.

**Strengths:**

Strengths are same as the summary and contributions.

**Additional Feedback:**

.

**Clarity:**

This paper is well-written.

The contributions and novelty are clear, compared to other benchmark dataset.

**Documentation:**

There is no URL for data access.

The authors said that the dataset will be released under the Apache License 2.0.

**Ethics:**

.

**Limitations:**

The authors did not provide their potential limitations and no discussion.

**Opportunities For Improvement:**

There is no information to download the proposed dataset in this document.

**Relation To Prior Work:**

It is clearly discussed in sections 1 and 2.
Tables and figures shows the difference from previous datasets in terms of various factors such as the number of classes/videos/tracks/boxes/frames, resolution, video length, etc.

**Summary And Contributions:**

Its contributions are well-summarized in introduction:
1) Large-scale open-vocabulary multi-object tracking benchmark (1,973 videos, 1,048 categories, and 637,608 bounding box annotations)
2) A diverse range of object categories: a total of 1,048 classes
3) Higher annotation density: a higher annotation frame rate (5-30 fps), compared to the recent related dataset OV-TAO-val (1 fps) for more detailed and comprehensive evaluations.

---

> ### Author Rebuttal · Authors · 2024-08-17
>
> Thanks for acknowledging our work and the inspirational questions. We provide a detailed response to your questions below.
>
> > 1. Computational Demands: it requires significant computational resources for training and evaluation.
>
> As discussed in the main paper, similar to the previous OV-TAO-val/OV-TAO-test, OVT-B serves as a comprehensive testing set. This way, it will not increase the computational resources for training. As a large-scale testing dataset, we acknowledge that the evaluation on OVT-B requires more computation. However, it seems natural that the computation goes up when evaluating on a larger benchmark for a more comprehensive evaluation.
>
> Inspired by the reviewer, we plan to design several protocols on OVT-B for the users to select if they want a quick and simple test.
>
> - Protocol I: Sub-set testing. The whole benchmark can be divided into different sub-sets, according to the dataset attributes (as discussed in Section 3.4), e.g., the object size, track length, or the video attributes of fast motion, occlusion, etc. This way, the evaluation can be implemented on the sub-set containing the videos with the concerned attributes.
>
> - Protocol II: Time sampling testing. As a video task, the evaluation can be also implemented after a temporal sampling, e.g., 1 fps, 5 fps (1 or 5 sampling frames per second). Such testing can reduce the testing computation while maintaining the dataset diversity.
>
> - Protocol III: Sub-task testing. The open-vocabulary multi-object tracking has three sub-tasks, i.e., object localization, classification, and tracking. For example, if the method aims at testing the tracking ability, it can directly use the provided detection results as input, which can reduce the testing time.
>
> We are going to enrich the testing protocols, in the final version of our paper (or supplementary material), for gradually perfecting the proposed OVT-B.
>
> > 2. Low Resolution: The average resolution is lower than that of the existing OV-TAO-val dataset.
>
> As shown in the table below, we provide the statistical magnitude, including the mean, median, and mode of the image resolution of OV-TAO-val and the proposed OVT-B. We can see that the average resolution of OVT-B (724 p) is just slightly lower than that of OV-TAO-val (788 p). Also, both the median and mode of the resolution in OVT-B are the same as those of OV-TAO-val. From the statistics, we find that, in terms of the image resolution, OVT-B is comparable with OV-TAO-val.
>
> |            | mean       | median     | mode       |
> |------------|------------|------------|------------|
> | OVT-B      | 1216 x 724 | 1280 x 720 | 1280 x 720 |
> | OV-TAO-val | 1308 x 788 | 1280 x 720 | 1280 x 720 |
>
> > 3. There is no information to download the proposed dataset in this document.
>
> We have added the download link of the proposed benchmark on the GitHub page. We also promise to continuously update more information on it in the future. Please see the below link for details.
> https://github.com/Coo1Sea/OVT-B-Dataset

---

### Official Review · Reviewer_RuFG · 2024-07-25
**OVT-B: A New Large-Scale Benchmark for Open-Vocabulary Multi-Object Tracking**

**Rating:** 8
**Confidence:** 3
**Correctness:** Good correctness.
**Clarity:** Yes

**Review:**

The paper is well-structured, with clear objectives, methodology, and results. The work is of good quality, demonstrated by the meticulous construction of the OVT-B dataset and the development of the OVTrack+ method. The authors have paid attention to detail in terms of dataset diversity, annotation accuracy, and the integration of motion features into the tracking algorithm.

**Strengths:**

1. The paper makes a significant contribution by introducing OVT-B, a large-scale benchmark that addresses a current gap in open-vocabulary multi-object tracking research. With its extensive dataset, it provides the research community with a robust platform for developing and evaluating new tracking methods, which is highly relevant to advancing AI in understanding and interacting with novel objects in video content.
2. The paper demonstrates a thorough approach to creating the benchmark, including careful curation and annotation of the dataset, and the development of OVTrack+, which shows improved performance over existing methods.

**Additional Feedback:**

No additional comments at this time.

**Documentation:**

Yes

**Ethics:**

No. The authors paid careful attention to the construction of their datasets. But they should claim their license for further usage of their data.

**Limitations:**

The authors adequately addressed the limitations and potential negative societal impact of their work.

**Opportunities For Improvement:**

This article generally doesn't have too many issues. What the reviewer is curious about is whether the authors can provide some insightful analysis on the current state of the open-set object tracking field based on the proposed dataset. Additionally, in 2D open-set detection, the definition of "open-set" is somewhat inconsistent. If the division is still based on base classes and novel classes, would this limit the development of the field for more real-world scenarios?

**Relation To Prior Work:**

Yes, there is clearly discussed.

**Summary And Contributions:**

The paper introduces OVT-B, a novel large-scale benchmark for open-vocabulary multi-object tracking, featuring 1,048 object categories and 1,973 videos with extensive bounding box annotations. It also presents OVTrack+, an enhanced baseline method that integrates motion features to improve tracking performance over its predecessor, OVTrack. The paper demonstrates the effectiveness of the new benchmark and method through extensive experiments and comparisons with existing state-of-the-art techniques. And it provides a solid data foundation for subsequent research on open-set object tracking.

---

> ### Author Rebuttal · Authors · 2024-08-17
>
> Thanks for your approval of this work and the questions to inspire us to think more deeply. We provide a detailed response to your questions below.
>
> > 1. What the reviewer is curious about is whether the authors can provide some insightful analysis on the current state of the open-set object tracking field based on the proposed dataset.
>
> Based on the results of the current state of the object tracking methods on the proposed dataset, we have some thoughts in the following.
> 1. As shown in Table 2 in the main paper, we can clearly see that the object classification results, especially for the novel class, are very low. This demonstrates that this is a very challenging problem having lots of room for improvement. So, how to improve the novel-class object classification ability of the open-set object tracking method, is a difficult problem worthy of further research.
>
> 2. In terms of the methods, we find that the current methods for open-set object tracking can not handle the performance balance among different sub-tasks. We think that the three sub-tasks, i.e., localization, classification, and tracking could be complementary to each other. For example, on the one hand, the correct tracking results can help the classification task, i.e., the predicted object category should be consistent along a trajectory. On the other hand, the object classification results can also help the tracking, where the object category can be used as a cue for temporal object association during tracking.
>
> 3. In classical multi-object tracking, the method can be divided into two categories, i.e., tracking-by-detection methods, and joint embedding of tracking and detection based methods. Classical MOT does not require classification during detection. In the open-vocabulary setting, the detection task is more challenging requiring open-class recognition ability.
> This way, from our point of view, the tracking-by-detection methods would be the mainstream framework in the near future. This is because the joint feature embedding for three different tasks is very challenging.
> As discussed in the above (second point), we think that using the results from different tasks to complement each other may be a better solution at present.
> We also hope to see the first effective joint embedding based method for OVMOT.
>
> 4. Finally, in terms of the evaluation metric for OVMOT, the existing overall metric TETA directly calculates the average of the localization, classification, and tracking accuracy rates. Considering the difficulty imbalance among the different sub-tasks, a new metric for more reasonable evaluation may be required.
>
> Thanks for your comment, which inspires us to think about the research direction of OVMOT in the future study.
>
> > 2. Additionally, in 2D open-set detection, the definition of "open-set" is somewhat inconsistent. If the division is still based on base classes and novel classes, would this limit the development of the field for more real-world scenarios?
>
> As discussed in the previous works [A][B][C] for 2D open-set detection or recognition, the open-set setting divides the object categories into the known (appearing in the training set) and unknown (not appearing in the training set) categories. The open-set detection methods aim to identify the unknown objects (for example ‘elephant’ [A]) with the label of ‘unknown’ during testing.
>
> In the open-vocabulary setting of this work, it also divides the object categories into two parts, base classes, and novel classes, similar to the known and unknown classes in open-set detection.
> Differently, it also requires the specific class of the novel objects, i.e., predicting the ‘elephant’ in the previous example with the label of ‘elephant’ during testing. This setting becomes more practical in real-world scenarios since we can recognize all the object categories (ideally) based on the training in some base classes. In particular, OVT-B dataset contains 1,048 categories (base+novel), which almost cover the common object categories in daily life. So the application of it can promote the development of the field of object tracking for more real-world scenarios.
>
> Certainly, this setting and benchmark also bring big new challenges for the methods. We hope these challenges can be gradually addressed with advanced studies, e.g., the latest multi-modal models, and large language models.
>
> [A] A. R. Dhamija, M. Günther, J. Ventura and T. E. Boult, The Overlooked Elephant of Object Detection: Open Set, IEEE Winter Conference on Applications of Computer Vision, 2020.
>
> [B] C. Gao, J. Hao and Y. Guo, OSDet: Towards Open-Set Object Detection, 2023 International Joint Conference on Neural Networks, 2023.
>
> [C] C. Geng, S. -J. Huang and S. Chen, Recent Advances in Open Set Recognition: A Survey, in IEEE Transactions on Pattern Analysis and Machine Intelligence, vol. 43, no. 10, pp. 3614-3631, 2021.

---

### Official Review · Reviewer_UMfb · 2024-07-30

**Rating:** 6
**Confidence:** 4
**Correctness:** The construction of dataset and bench…
**Clarity:** 1. The quality of the figures used in…

**Review:**

This paper conducts a new large scale benchmark for open-vocabulary multi-object tracking namely OVT-B. Comparing to the recent benchmark OV-TAO, the proposed dataset contains a large number of categories (1048 v.s. 330) and annotated boxes (673K v.s. 279K). Then the authors conduct experiments with recent methods and provide a new baseline OVTrack+.

Pros:
1. Providing a new large scale dataset with a large number of categories and annotated results.
2. Detailed illustration about the dataset collection, annotation and distribution.
3. Providing several baselines methods on the datasets with experimental analysis.

Cons:
1. The authors provide a link about the dataset on GitHub. However, the only thing in the GitHub repo is a README without any download links or data samples.
2. Missing implementation details about the providing baseline methods. What is the detector used in the experiments? What is the performances of the used detector in the proposed dataset?
3. The providing baselines methods are all tracking-by-detection methods. What about joint tracking and detection methods such as MOTR [1]?
4. The presentation should be improved. Please refer to the Clarity for detailed.

[1] MOTR: End-to-End Multiple-Object Tracking with Transformer. ECCV 2022.

**Strengths:**

Please refer to the Pros in Review.

**Additional Feedback:**

Please provide a full link describe the detailed organization of dataset and illustrations of data samples.

**Documentation:**

The authors describe the data collection and annotation in the main paper. However, they do not provide any data samples and download link to explore the dataset.

**Ethics:**

There are not ethical concerns in the submission.

**Limitations:**

The authors discuss the limitations in the paper. The proposed limitations are focus on the training methods and framework for OV-MOT. In the paper, the authors do not address the limitations as the main contribution of this paper is to provide a new benchmark.

**Opportunities For Improvement:**

Please refer to the Cons in Review.

**Relation To Prior Work:**

The authors discuss the differences between the proposed benchmark and prior work OV-TAO.

**Summary And Contributions:**

This paper focuses on the topic of open-vocabulary multi-object tracking (OVT) and proposes a new large-scale benchmark named Open-Vocabulary Tracking Benchmark (OVT-B) for OVT. The authors first analyze the drawbacks of the recent popular OVT dataset OV-TAO: small number of categories and sparse annotation rate. They then describe the data collection and annotation pipeline of the proposed dataset OVT-B, which contains 1937 videos and 637,608 annotated objects from 1048 categories. The authors also describe the distribution of the dataset. Finally, they conduct experiments with recent state-of-the-art tracking methods and provide in-depth experimental analysis.

---

> ### Author Rebuttal · Authors · 2024-08-17
>
> Thanks for acknowledging our work and the constructive thoughts. We provide a detailed response to your questions below.
> > 1. The authors provide a link about the dataset on GitHub. However, the only thing in the GitHub repo is a README without any download links or data samples. Please provide a full link describe the detailed organization of dataset and illustrations of data samples.
>
> Thanks for your reminder. We have added the download link of the proposed benchmark on the GitHub page, which also provides the detailed organization of the dataset and illustrations of data samples. We promise to continuously update more information on it in the future. Please see the below link for details.
> https://github.com/Coo1Sea/OVT-B-Dataset
>
> > 2. Missing implementation details about the providing baseline methods. What is the detector used in the experiments? What is the performances of the used detector in the proposed dataset?
>
> Following previous work, i.e., our baseline method OVTrack [A], we also use the detector proposed in ViLD [B] in the experiments. Note that, for fair comparison, all the comparison methods in Tables 1 and 2 use the same detector.
> To show the performance of the detection, we first use the classical average precision (AP50) score as a metric. For fine-grained analysis, we also use two metrics [C], i.e., the localization and classification, respectively.
> LocA is used to evaluate object localization, in which a prediction box that has an IoU higher than a localization threshold of 0.7 with a ground truth box is taken as a true positive.
> ClsA is used to evaluate the object classification, which is an independent score only reflecting the pure performance of the category recognition.
> The performance is provided in the following table.
>
> |      | All   |       |       | Base  |       |      | Novel |      |
> |------|-------|-------|-------|-------|-------|------|-------|------|
> | AP   | LocA  | ClsA  | AP    | LocA  | ClsA  | AP   | LocA  | ClsA |
> | 7.0% | 62.0% | 11.3% | 7.9 % | 61.6% | 13.2% | 6.1% | 62.5% | 9.4% |
>
> We can see that the detection AP score is low. This is because the object classification is very difficult.
> Specifically, we can see that the localization performance in object detection is acceptable.
> However, the object classification performance is poor.
> The main reasons are 1) the number of categories in the proposed dataset is very large, which makes the classification task hard.
> 2) The novel classes do not appear during training, thus the classification of them is more difficult.
> From the above analysis, we can see that object recognition (classification) is a very difficult task in OVMOT, on OVT-B, which has great space for improvement in the future.
>
> [A] Li S, Fischer T, Ke L, et al. OVTrack: Open-vocabulary multiple object tracking. IEEE/CVF Conference on Computer Vision and Pattern Recognition, 2023.
>
> [B] Gu X, Lin T Y, Kuo W, et al. Open-vocabulary object detection via vision and language knowledge distillation. The International Conference on Learning Representations, 2022.
>
> [C] Li S, Danelljan M, Ding H, et al. Tracking everything in the wild. European Conference on Computer Vision, 2022.
>
> > 3. The providing baselines methods are all tracking-by-detection methods. What about joint tracking and detection methods such as MOTR [1]?
>
> Previous MOT mainly focuses on specific targets, i.e., human [D, E] or vehicle [F], which also does not require object classification during detection. Differently, in the open-vocabulary MOT setting, the detector should be class-agnostic, followed by a classifier with the open-vocabulary recognition ability. This makes the joint tracking and detection methods unsuitable. For example, MOTR is end-to-end trained on the datasets of MOT 15 [D], MOT 17 [E], and DanceTrack [G], all of which contain only the object of humans. This makes MOTR able to only identify one object category of humans during the testing on the proposed OVT-B, which is inapplicable.
>
> This way, the tracking-by-detection method is the mainstream framework in the OVMOT task, in which the class-agnostic detector can be pre-trained on the large-scale image dataset, e.g., LVIS, with various categories.
> Thanks for the reviewer’s comment, which inspires us to consider the new joint tracking and detection method for OVMOT.
> Also, for the model training, a large-scale video tracking training dataset with various categories is also worth building.
>
> [D] Leal-Taixé L, Milan A, Reid I, et al. Motchallenge 2015: Towards a benchmark for multi-target tracking. arXiv:1504.01942, 2015.
>
> [E] Milan A, Leal-Taixé L, Reid I, et al. MOT16: A Benchmark for Multi-Object Tracking. arXiv:1603.00831
>
> [F] Geiger A, Lenz P, Urtasun R. Are we ready for autonomous driving? The KITTI vision benchmark suite. IEEE Conference on Computer Vision and Pattern Recognition, 2012.
>
> [G] Sun P, Cao J, Jiang Y, et al. DanceTrack: Multi-Object Tracking in Uniform Appearance and Diverse Motion. IEEE/CVF Conference on Computer Vision and Pattern Recognition, 2022.
>
> > 4. The presentation should be improved. The quality of the figures used in this paper needs improvement. At a minimum, the bitmap images should be replaced with vector graphics, or higher resolution bitmaps should be used. Additionally, related figures describing different aspects should be organized using the subfigure or minipage environment in LaTeX. Do not use figure like Figure 6 in the paper.
>
> Thanks for your careful review. We will proofread our manuscript in the final version. Also, as per your suggestion, we have updated the figures with clearer visualization. We also use the subfigure to show the figure with different aspects, including Figures 4 and 6. Please see the attached PDF file for these figures.

---

### Decision · Program_Chairs · 2024-09-26

**Decision:**

Accept (Poster)

**Comment:**

This paper presents a new large-scale benchmark named Open-Vocabulary Tracking Benchmark (OVT-B) for OVT. The paper describes the data collection and annotation pipeline of the proposed dataset OVT-B, which contains 1937 videos and 637,608 annotated objects from 1048 categories. The paper also describes the distribution of the dataset. Experiments with recent state-of-the-art tracking methods are also provided with detailed experimental analysis. This paper also analyzes the drawbacks of the recent popular OVT dataset OV-TAO: small number of categories and sparse annotation rate.

All of the three reviewers are positive to this paper and recommend for an acceptance. The rebuttal partially addresses some of the reviewer concerns. AC agree with reviewers to accept this paper.